# Deconvolution Analysis of the Non-Ionic Iomeprol, Iobitridol and Iodixanol Contrast Media-Treated Human Whole Blood Thermograms: A Comparative Study

**DOI:** 10.3390/diagnostics13152523

**Published:** 2023-07-28

**Authors:** Elek Telek, Zoltán Ujfalusi, Miklós Nyitrai, Péter Bogner, András Lukács, Tímea Németh, Gabriella Hild, Gábor Hild

**Affiliations:** 1Department of Biophysics, Medical School, University of Pécs, Szigeti Str. 12, H-7624 Pécs, Hungary; elek.telek@aok.pte.hu (E.T.); zoltan.ujfalusi@aok.pte.hu (Z.U.); miklos.nyitrai@aok.pte.hu (M.N.); andras.lukacs@aok.pte.hu (A.L.); 2Szentágothai Research Center, Ifjúság Str. 34, H-7624 Pécs, Hungary; 3MTA-PTE Nuclear-Mitochondrial Interactions Research Group, Szigeti Str. 12, H-7624 Pécs, Hungary; 4Department of Medical Imaging, Clinical Centre, University of Pécs, Ifjúság Str. 13, H-7624 Pécs, Hungary; bogner.peter@pte.hu; 5Languages for Biomedical Purposes and Communication, Medical School, University of Pécs, Szigeti Str. 12, H-7624 Pécs, Hungary; timea.nemeth@aok.pte.hu (T.N.); gabriella.hild@aok.pte.hu (G.H.)

**Keywords:** human blood, plasma, serum, contrast media, Iomeprol, Iobitridol, Iodixanol, calorimetry, deconvolution

## Abstract

To study the effect of non-ionic contrast media on anticoagulated and non-anticoagulated human whole blood samples, calorimetric measurements were performed. The anticoagulated plasma showed the greatest fall in the total *ΔH* after Iodixanol treatment. The plasma-free erythrocytes revealed a pronounced shift in the *T_max_* and a decrease in the *ΔH* of hemoglobin and transferrin. The total *ΔH* of Iodixanol treatment showed the highest decline, while Iomeprol and Iobitridol had fewer adverse effects. Similarly, the non-anticoagulated samples revealed a decrease both in the *T_max_* and the *ΔH* of albumin and immunoglobulin-specific transitions. The total *ΔH* showed that Iodixanol had more influence on the serum. The serum-free erythrocyte samples resulted in a significant drop in the *T_max_* of erythrocyte and transferrin (~5–6 °C). The *ΔH* of deconvolved hemoglobin and transferrin decreased considerably; however, the *ΔH* of albumin increased. Surprisingly, compared to Iomeprol and Iobitridol treatments, the total *ΔH* of Iodixanol was less pronounced in the non-anticoagulated erythrocyte samples. In sum, each non-ionic contrast medium affected the thermal stability of anticoagulated and non-anticoagulated erythrocyte proteins. Interestingly, Iodixanol treatment caused more significant effects. These findings suggest that conformational changes in blood components can occur, which can potentially lead to the increased prevalence of cardiovascular dysfunctions and blood clotting.

## 1. Introduction

Contrast media are routinely used for contrast enhancement of human biological structures in medical imaging, such as Computed Tomography (CT), X-ray and Magnetic Resonance Imaging (MRI) [1,2,3]. Since several tissues have similar attenuation properties, highly attenuating contrast agents are required to visualize different human organs efficiently [4]. Intravenous or oral administration of contrast media results in physical interaction with either the intra- or extracellular side of cells. These interactions may affect the structure and function of intracellular proteins.

Iodine-based contrast media are water-soluble and can be divided into two groups: ionic and non-ionic compounds. Ionic contrast media have more adverse effects due to the high osmolality. These severe adverse effects include thyroid dysfunctions, hypersensitivity reactions and contrast agent-induced nephropathy (CIN). At the molecular level, these conditions can be related to chemotoxicity (e.g., binding of proteins), osmotoxicity (inhibiting the transportation of water molecules through membranes) and ion toxicity of the contrast media (ionic compounds might be able to change cellular functions) [4]. Anticoagulant effects, which have been described earlier [5], are more characteristic of ionic than non-ionic contrast media. Although one randomized trial found no significant difference in the procoagulation effects between non-ionic and ionic contrast media [6], most studies showed that procoagulant effects were more typical for non-ionic contrast media [7]. Therefore, using non-ionic contrast media is more likely to lead to thrombus formation and adverse cardiac dysfunctions than applying ionic ones. According to a recent meta-analysis, acute adverse reactions can occasionally occur while using a newly developed non-ionic contrast media [8]. Having low osmolality, several non-ionic contrast media can change the deformability of erythrocytes. Red blood cells have a crucial role in the maintenance of physiological blood flow and systemic circulation; therefore, the normal shape and structure of erythrocytes are essential [9].

Nowadays, non-ionic contrast media are used more frequently because they lead to fewer adverse effects due to their decreased osmolality, and their application is safer compared to the ionic ones [10,11]. In spite of their lower osmolality, adverse effects can still occur [12,13,14]. Keeping the contrast agent concentration as low as possible and using multidetector-row CT scanners can still provide reasonable contrast and proper quality data for particular CT examinations [15].

In our study, we investigated the effects of three non-ionic, iodine-containing contrast media, Iomeprol, Iobitridol and Iodixanol, on human whole blood samples. Iomeprol (Iomeron^®^ 400 mg I/mL) is a non-ionic, monomeric, water-soluble X-ray contrast agent. It consists of three covalently bound iodine atoms and has low osmolality, chemotoxicity and viscosity. The osmolality and viscosity of Iomeprol is lower compared to other non-ionic contrast media [16]. Iomeprol does not interact substantially with serum proteins and is not metabolized as quickly as it is eliminated through the urinary tract [17]. Although the pharmacodynamic effects of Iomeprol are not significant in humans, metabolic changes were detected in an animal model, such as high bilirubin, gamma glutamic transferase and glucose level. Increased protein and enzymuria can also occur transiently [18]. Clinical examinations of Iomeprol administration suggest that it is well tolerated by patients [19].

Similarly to Iomeprol, Iobitridol (Xenetix^®^ 350 mg I/mL) is a monomer, non-ionic radiopaque, and highly water-soluble contrast agent with low osmolality and viscosity [20]. It is mainly applied in CT examinations. Iobitridol molecules contain three iodine atoms bound to a benzene ring. Chemotoxical effects have been studied through binding experiments between Iobitridol and porcine pancreatic elastase. Prange and his coworkers found that the administration of Iobitridol at 0.5 M concentration caused a low number of interactions, concluding that Iobitridol had a low activity compared to other contrast media used in their experiments [21]. According to comparative studies, Iobitridol is well tolerated by patients. The statistical analysis of a post-marketing study also confirmed that the ratio of the adverse effects was less than 1%, and the rate of severe adverse reactions was below 0.05% without any casualties [22]. Other studies also confirmed that Iobitridol was still one of the safest iodinated contrast media, causing the least serious adverse effects on patients [23,24].

Iodixanol (Visipaque^®^ 320 mg I/mL) is a non-ionic, hydrophilic and dimeric iodine-based contrast agent with low osmolality and viscosity [25]. In Iodixanol molecules, three iodine atoms are bound to each benzene ring in a dimeric structure. The studies investigating the chemotoxicity of Iodixanol showed no change in the platelet function of human platelet samples under in vitro conditions [26]. In the previously mentioned experiments, an antiaggregant activity of Iodixanol was found, suggesting that the antiaggregant activities of some medications can be changed [27]. In addition, Iodixanol and other iodine-based contrast agents have cytotoxic effects on particular cancer cell lines [28]. Severe nephrotoxicity [29], as well as microcirculatory dysfunctions, can also be assigned to Iodixanol for developing contrast-induced acute kidney failure [30]. Administering low-concentration Iodixanol (270 mg I/mL) and using CT with decreased peak voltage (100 kVp) provides high resolution and good image quality with repetitive reconstructions while minimizing the radiation burden. Studies comparing the effect of Iodixanol to other iodinated contrast media showed similar or less adverse reactions [29,31,32]. Iodixanol can be applied as an active material of Visipaque in the therapeutic treatment of allergic rhinoconjunctivitis [33]. It can also contribute to the higher purity and longer lifetime of human islet cultures [34]. Although data are available regarding the effects of non-ionic contrast media on blood samples, their effects on the thermal stability of blood proteins have not yet been investigated. The aim of our experiments was to study the effect of Iomeprol, Iobitridol and Iodixanol on human whole blood plasma, serum, and erythrocytes, using differential scanning calorimetry (DSC).

## 2. Materials and Methods

### 2.1. Preparation of Human Blood Samples

Blood samples were collected from healthy volunteers by healthcare professionals and were used for plasma and serum preparation. The applied procedures were approved by the Local Ethical Committee of the UP MS (Certificate No. 8549-PTE2020). 

The blood plasma was prepared from whole blood in anticoagulant-treated Vacutainer^®^ tubes with light blue tops (BD Vacutainer^®^ Citrate Tubes with 3.2% buffered sodium citrate solution for routine coagulation studies) and then centrifugated for 15 min at 4000× *g* at 4 °C. The plasma was gently pipetted off from the top right after centrifugation, and both the plasma and the pellet were stored on ice until the measurements. The blood samples collected in red topped Vacutainer^®^ plastic serum tubes (BD Vacutainer^®^ Plus Plastic Serum Tubes with spray-coated silica for routine blood screening and serum determinations) were left to clot for 10–15 min, and the clotted blood was centrifugated for 15 min at 4000× *g* at 4 °C. The serum supernatant was carefully transferred from the top immediately after the centrifugation, and the serum and pellet were stored on ice until use. 

### 2.2. Treatment with Contrast Media

The anticoagulated and non-anticoagulated blood samples were treated separately with Iomeron, Xenetix and Visipaque contrast agents before the calorimetric measurements. 

Iomeron^®^ 300 solution contains 61.24% (*m*/*v*) Iomeprol (N, N′-bis(2, 3-dihydroxypropyl)-5[(hydroxyacetyl)methylamino]-2, 4, 6-triiodo-1, 3-benzenedicarboxamide) as an active agent. Xenetix^®^ 350 mg I/mL solution consists of 383.90 g Iobitridol (N,N′-Bis(2,3-dihydroxypropyl)-5-(2-(hydroxymethyl)hydracrylamido)-2,4,6-triiodo-N,N′-dimethylisophthalamide) (corresponds to 175.0 g iodine) as active material. Visipaque^®^ 320 mg I/mL stock solution consists of 326.0 g Iodixanol (5-[acetyl-[3-[acetyl-[3,5-bis(2,3-dihydroxypropylcarbamoyl)-2,4,6-triiodo-phenyl]amino]-2-hydroxy-propyl]amino]-N,N′-bis(2,3dihydroxypropyl)-2,4,6-triiodo-benzene-1,3-dicarboxamide) (equivalent to 160.0 g iodine) as an active material. Contrast media were applied 10 min before each measurement by gently mixing them with the blood samples in a concentration of 40 mM, which is twice as high as the concentration typically given to patients (20 mM). The aim of our study was to compare the effects of the applied contrast media on blood plasma, serum, and erythrocyte components.

### 2.3. Differential Scanning Calorimetry (DSC) Measurements

DSC is a comprehensive method to measure the thermodynamic properties of biological and clinical samples [35,36,37,38,39,40]. In our experiments, the thermal denaturation of anticoagulated and non-anticoagulated blood samples was studied with a SETARAM µDSC-III calorimeter. Each DSC measurement was carried out at a heating rate of 0.3 K·min^−1^ between 20–100 °C. During the sample preparation (appropriate mixture of blood sample and contrast agent), 850 µL of sample or normal saline (reference) was pipetted in conventional Hastelloy batch vessels (*V_max_* = ~1 mL) for the measurements. The sample and reference vessels were balanced with a precision of ±0.05 mg; therefore, correction was not needed with the heat capacity of the vessels. The second denaturation procedure was used for the baseline correction of the data. At the beginning of measurements, the Hastelloy batch vessel containing the given sample is at atmospheric pressure. During measurements, the pressure can increase inside the vessel, leading to approximately 2 bars (202.6 kPa) at the end of the heating cycle. At lower temperatures, denaturation of blood components can occur, suggesting that the increased pressure has no significant effect on the thermal transition of proteins. 

### 2.4. Deconvolution of DSC Thermograms

Anticoagulated and non-anticoagulated blood plasma, serum and cellular elements of human whole blood consist of several proteins [41,42,43]; therefore, the deconvolution analyses were performed using OriginLab Origin^®^2020 software. To quantify the thermodynamic properties of our samples, we determined the melting temperatures (*T_m_*). The *T_m_* value of the denaturation indicates the peak temperature of the calorimetric thermal curve, where 50% of the sample is unfolded. The value of the *T_m_* is a key indicator of the thermodynamic stability of proteins. The greater the *T_m_* value, the more stable the structure of the protein is [44]. *T_m_* value is used when the protein is highly pure, and the denaturation mechanism is supposed to be a single-step transition. In our measurements, the fractionated blood samples contained multiple protein components. Therefore, *T_max_* was used as an indicator of the thermodynamic stability of the complex system. The normalized enthalpy change (*ΔH*) was calculated using the area under the deconvolved thermal curves.

## 3. Results

In order to compare the effects of Iomeprol, Iodibitridol and Iodixanol on the thermodynamic properties of anticoagulated and non-anticoagulated human whole blood samples (plasma, serum and erythrocytes), DSC measurements were performed. 

### 3.1. Deconvolution of the Thermal Curves of Anticoagulated Blood Plasma and Cellular Components

#### 3.1.1. Analysis of Anticoagulated Blood Plasma

The deconvolution analysis of blood plasma thermal curves revealed a major albumin peak and three minor transition peaks in the immunoglobulin-specific shoulder both in the absence and the presence of the contrast agents. The melting temperature of the albumin transition (*T_max1(c)_*) was 62.9 °C, which is in accordance with previous findings [35,41,45,46,47,48]. The presence of 40 mM Iomeprol, Iobitridol and Iodixanol did not induce a significant change in the *T_max_* of albumin (Figure 1, Table 1). According to previous results, immunoglobulin-specific proteins can be observed in the range of ~68–75 °C [46,49]. In our experiments, the control *T_max_* of the deconvolved immunoglobulin-specific protein curves appeared at *T_max2(c)_ =* 67.3 °C, *T_max3(c)_ =* 71.2 °C and *T_max4(c)_ =* 75.4 °C, respectively; while the treatment with 40 mM Iomeprol, Iobitridol and Iodixanol produced a negligible change in these *T_max_* values (Figure 1, Table 1). 

No significant change could be observed in the enthalpy of the main transition peak in the presence of the contrast media (Figure 2a, Table 2). In addition, the *ΔH* of the deconvolved immunoglobulin-specific curves were *ΔH_2(c_*_)_ = 0.04 J·g^−1^, *ΔH_3(c_*_)_ = 0.02 J·g^−1^ and *ΔH_4(c_*_)_ = 0.02 J·g^−1^, respectively. Both the 40 mM Iomeprol and the Iodixanol treatments resulted in a considerable decrease in the enthalpy of each deconvolved immunoglobulin curve; however, no significant change was found during the treatment using 40 mM Iobitridol (Figure 2a, Table 2). Surprisingly, the strongest effect developed after the Iodixanol treatment, where the enthalpy of the deconvolved immunoglobulin curves, mainly peak 3 and 4, fell drastically (*ΔH_2(v_*_)_ = 0.03 J·g^−1^, *ΔH_3(v_*_)_ = 0.01 J·g^−1^, *ΔH_4(v_*_)_ = 0.005 J·g^−1^) (Figure 2a, Table 2). In line with these results, the change in the total enthalpy of the plasma samples was *ΔH_total(v)_* = 0.14 J·g^−1^ (~23% decrease), whereas the control value of *ΔH_total_* was 0.18 J·g^−1^ (Figure 2b, Table 3), indicating that Iodixanol could cause more harmful effects mainly on immunoglobulins, compared to the other contrast materials applied in our study.

#### 3.1.2. Analysis of Anticoagulated Cellular Components

We also investigated the plasma-free erythrocyte samples in the absence and presence of 40 mM Iomeprol, Iobitridol and Iodixanol. The analysis revealed that besides the hemoglobin-specific major peak [50], two thermal curves could be deconvolved in the second minor transition (Figure 3a). Previous studies have suggested that a peak at approximately 80 °C may be attributed to a predominantly helical erythrocyte component [51], while the thermal transition at approximately ~82 °C is probably attributed to transferrin depending on the number of bound irons [52]. In line with previous findings, the deconvolved two minor peaks of the second thermal transition between 73–88 °C might be helical erythrocyte proteins and transferrin. The control melting temperature of the major hemoglobin peak 1 was *T_max1(c)_* = 70.5 °C, which is similar to previous findings (72 °C) and may be due to the different scanning speed and setting parameters [53]. The *T_max_* of the deconvolved control minor transitions were *T_max2(c)_* = 76.7 °C and *T_max2(c)_* = 81.7 °C, respectively. The *T_max_* of the major hemoglobin peak decreased the most with the 40 mM Iodixanol treatment (*T_max1(v)_* = 68.7°C), while the effect of Iomeprol and Iobitridol was less pronounced (Figure 3, Table 4). In the absence of contrast media, the *T_max_* of the deconvolved minor curves was 76.7 °C (*T_max2(c)_*) and 81.7 °C (*T_max3(c)_*). The addition of 40 mM Iomeprol and Iobitridol slightly decreased *T_max_*, while the use of 40 mM Iodixanol caused the most significant thermal shift (*T_max2(v)_* = 74.9 °C and *T_max3(v)_* = 78.5 °C) (Figure 3, Table 4). The DSC measurement also showed a very shallow endothermic transition assigned to membrane components of the erythrocytes [50] between 47–53 °C. This transition was not influenced by the contrast media and could not be deconvolved. The melting temperature values revealed that Iodixanol had a stronger negative effect on the thermodynamic stability of anticoagulated erythrocyte proteins compared to Iomeprol and Iobitridol. 

The deconvolved enthalpy change of the major hemoglobin curve was *ΔH_1(c)_* = 0.862 J·g^−1^. It did not change considerably in the presence of 40 mM Iomeprol and Iobitridol. The greatest decrease could be observed after the treatment with 40 mM Iodixanol (*ΔH_1(v)_* = 0.747 J·g^−1^), which resulted in a 14% fall in the *ΔH* (Figure 4a, Table 5). The enthalpy change of the control minor transitions were *ΔH_2(c)_* = 0.06 J·g^−1^ and *ΔH_3(c)_* = 0.23 J·g^−1^, respectively. Interestingly, each treatment led to a significant drop in the *ΔH* of both minor transitions of anticoagulated erythrocyte samples (Figure 4a, Table 5). In line with this, the enthalpy changes of Iomeprol-, Iobitridol- and Iodixanol-treated second minor transition decreased with 51%, 49% and 37% (*ΔH_2(i)_*_,_ *ΔH_2(x)_*
_and_
*ΔH_2(v)_*), compared to the control (*ΔH_2(c)_*) value. Furthermore, the enthalpy change of the third minor peak showed a drastic fall after the treatment with Iomeprol (88%), Iobitridol (80%) and Iodixanol (84%) (Figure 4a, Table 5). The total change in the enthalpy was the highest (29%) in the Iodixanol-treated erythrocyte samples (*ΔH_total(v)_* = 0.82 J·g^−1^), compared to the Iomeprol (22%) and Iobitridol treatments (13%) (Figure 4b, Table 6). This indicates that the overall adverse effect of Iodixanol was more pronounced on the conformation of proteins of the anticoagulated erythrocyte samples. 

These results suggest that less energy is required to unfold the major hemoglobin peak in the case of Iodixanol treatment, resulting in more adverse effects compared to Iomeprol and Iobitridol. The enthalpy changes of the minor deconvolved curves showed a more drastic decrease, suggesting that Iomeprol, Iobitridol and Iodixanol all had a similarly strong adverse effect on the thermodynamic properties of anticoagulated erythrocyte components.

In sum, Iomeprol, Iobitridol and Iodixanol may primarily cause conformational changes in the minor transitional components, which can be attributed to the unknown transition and transferrin (peaks 2 and 3). These data suggest a stronger affinity of the contrast media to these proteins. Moreover, the major hemoglobin peak 1 was more influenced by Iodixanol. In summary, the total change in enthalpy decreased the most in the case of Iodixanol, indicating the most adverse effect on anticoagulated erythrocyte proteins.

### 3.2. Deconvolution of the Thermal Curves of Non-Anticoagulated Blood Serum and Cellular Elements 

#### 3.2.1. Analysis of Non-Anticoagulated Blood Serum

The deconvolution of blood serum revealed one major and three immunoglobulin-specific transition curves in the absence and presence of 40 mM contrast media. The control melting temperature of the major curve assigned to human serum albumin (HSA) [46,49] was *T_max1(c)_* = 63.4 °C, which decreased with 1.6 °C in the presence of Iomeprol and Iodixanol, whereas the Iobitridol treatment resulted in a lower decrease (1.1 °C) in the *T_max_* value (Figure 5, Table 7). 

The melting temperatures of the control immunoglobulin-specific deconvolved transition curves were *T_max2(c)_* = 67.6 °C, *T_max3(c)_* = 71.1 °C and *T_max4(c)_* = 75.1 °C, respectively (Figure 5, Table 7). The presence of 40 mM Iomeprol, Iobitridol and Iodixanol caused similarly small thermal shifts in the minor transitions. However, the treatment with Iodixanol resulted in a larger decrease in *T_max_* for peak 3 (1.1 °C) and peak 4 (1.6 °C), compared to Iomeprol and Iobitridol (Figure 5, Table 7). These results suggest that Iodixanol had a more negative effect on the conformation of these immunoglobulins. 

Regarding the enthalpy change of non-anticoagulated blood serum, the treatments with Iomeprol and Iodixanol were found to have a more significant impact than Iobitridol. In the presence of 40 mM Iomeprol, the *ΔH* of the deconvolved HSA major peak decreased by 12%. Compared to the control samples, the *ΔH* of the deconvolved HSA major peak fell more considerably (19%) following the Iodixanol treatment (Figure 6a, Table 8). Overall, the Iodixanol treatment produced the most significant effect on the major HSA peak. 

The deconvolution of the immunoglobulin-specific transitions revealed that Iomeprol, Iobitridol and Iodixanol impacted the second minor transition primarily (peak 3), decreasing the *ΔH* with 34%, 17% and 23%, respectively. Each contrast agent caused a change in the enthalpy of the third minor transition (peak 4). A considerable increase of the *ΔH* was observed in the presence of Iomeprol (28%), Iobitridol (46%) and Iodixanol (32%) (Figure 6a, Table 8).

The change in the total enthalpy was more pronounced, and almost all contrast media caused a considerable change under the area of the deconvolved transition curves (Figure 6b, Table 9). The presence of 40 mM Iodixanol decreased (14%) the total enthalpy the most (*ΔH_total_ =* 0.19 J·g^−1^), suggesting that Iodixanol had the strongest adverse effect on the thermodynamic properties of non-anticoagulated serum samples (Figure 6b, Table 9). Compared to the measurements on anticoagulated plasma, the *ΔH* of non-anticoagulated serum was less significant. The possible explanation is that the coagulation factors affect the affinity of contrast media to plasma proteins, thereby influencing thermal stability.

#### 3.2.2. Analysis of Non-Anticoagulated Blood Cellular Elements

In addition, we performed calorimetric experiments on non-anticoagulated erythrocyte samples in the absence and presence of 40 mM Iomeprol, Iobitridol and Iodixanol. The melting temperatures of erythrocyte control samples were *T_max1(c)_* = 69.6 °C, *T_max2(c)_* = 81.6 °C and *T_max3(c)_* = 85.6 °C (Figure 7, Table 10), which might be attributed to hemoglobin (~69°C), helical erythrocyte protein component (~82 °C) and transferrin (~85 °C), respectively [49]. In our study, the hemoglobin transition curve showed a minor decrease compared to the *T_max_* found in previous findings (~72 °C) [50], which is probably due to the difference in the scanning speed (in our experiments, 0.3 K·min^−1^ compared to 1 K·min^−1^). The presence of 40 mM Iomeprol, Iobitridol and Iodixanol caused no major changes in the *T_max_* of hemoglobin. In addition, each contrast medium produced a drastic thermal shift (~5–6 °C decrease) in the deconvolved minor transitions (peaks 2 and 3), which can be attributed to the presence of helical erythrocyte and transferrin (Figure 7a, Table 10).

Interestingly, the calculated enthalpy of deconvolved non-anticoagulated erythrocyte transition curves increased by 30%, 33% and 22% in the presence of 40 mM Iomeprol, Iobitridol and Iodixanol, respectively (Figure 8a, Table 11). In contrast with the major hemoglobin peak, 40 mM Iomeprol and Iobitridol significantly decreased the enthalpy of the helical erythrocyte (91%, 93%) and transferrin (73%, 80%) transition curves, respectively. Furthermore, 40 mM Iodixanol substantially reduced the enthalpy; however, the change was less adverse on the helical erythrocyte protein (75%) and transferrin (50%) compared to the Iomeprol and Iobitridol treatments (Figure 8a, Table 11). 

In line with this, the total enthalpy change was slightly less pronounced in the presence of Iodixanol (27%) compared to the Iomeprol and Iobitridol treatments (33% and 31%), respectively (Figure 8b, Table 12). These data suggest that Iomeprol and Iobitridol had a similarly strong adverse effect, and Iodixanol might have less influence on the thermal stability of helical erythrocytes and transferrin.

In sum, the applied contrast media had no major effect on the melting temperature of hemoglobin. However, a significant increase was observed in the enthalpy of Iomeprol and Iobitridol-treated samples, whereas the Iodixanol treatment revealed a less adverse effect. The change in the *T_max_* and *ΔH* resulted in a drastic fall in the helical erythrocyte and transferrin thermal transitions with each contrast medium, but it was less pronounced in the Iodixanol-treated samples, suggesting that Iomeprol and Iobitridol had more significant adverse effects. Iodixanol could not influence the helical erythrocyte and transferrin as strongly as Iomeprol and Iobitridol.

## 4. Discussion

Contrast media are widely used in routine medical imaging, such as CT, X-ray and MRI [1,2,3]. The administration of contrast media results in physical interaction with either the intra- or extracellular components of the cells and may influence the conformation and function of proteins, leading to the development of severe conditions, such as thyroid dysfunctions, hypersensitivity and contrast agent-induced nephropathy (CIN) [54,55]. To reduce the likelihood of interaction, non-ionic contrast media are being used more often nowadays. The use of non-ionic contrast media is expanding due to its lower osmolality, resulting in fewer harmful effects and greater safety compared to ionic contrast media [10,11]. However, adverse effects may still occur [12,13,14]. 

To study the effect of three non-ionic, iodine-containing contrast media, DSC measurements were carried out on anticoagulated and non-anticoagulated human whole blood samples using 40 mM of Iomeron, Xenetix or Visipaque. The melting temperature (*T_max_*) and the change in the enthalpy (*ΔH*) were determined in order to characterize the detailed thermodynamic effects of the contrast media on human whole blood components.

In the case of anticoagulated blood samples, 40 mM Iodixanol had a slightly stronger effect on the thermodynamic stability of the deconvolved erythrocyte transition curves (hemoglobin, helical erythrocyte component and transferrin deconvolved peaks) than on the plasma samples, compared to Iomeprol and Iobitridol (Figure 1, Figure 2, Figure 3 and Figure 4 and Table 1, Table 2, Table 3, Table 4, Table 5 and Table 6). Moreover, the change in the enthalpy revealed that Iodixanol decreased the thermodynamic properties of both the deconvolved plasma and erythrocyte samples to a greater extent than the other contrast agents. This indicates that less energy is required to unfold the structure of proteins. However, Iomeprol and Iobitridol have similarly adverse effects (Figure 1, Figure 2, Figure 3 and Figure 4 and Table 1, Table 2, Table 3, Table 4, Table 5 and Table 6). The total change in enthalpy also confirms that Iodixanol had the most significant adverse effect on both the anticoagulated plasma (*ΔH_total_* = ~23%) and erythrocyte protein samples (*ΔH_total_* = ~29%). However, Iomeprol and Iobitridol also caused considerable harmful effects (Figure 2b and Figure 4b, Table 3 and Table 6). 

The thermal denaturation analysis of non-anticoagulated blood serum samples indicated that 40 mM Iodixanol caused a slightly greater thermal shift in the deconvolved serum transition curves of HSA and immunoglobulin-specific proteins than Iomeprol and Iobitridol (Figure 5, Table 7). Consistent with this, Iodixanol treatment resulted in the largest decrease in the enthalpy change of the deconvolved HSA transition peak 1. In contrast, no significant enthalpy change was observed in the deconvolved transitions of immunoglobulin-specific curves (peaks 2, 3, and 4) using either contrast medium (Figure 6a, Table 8).

Overall, the total enthalpy change indicates that Iodixanol had the strongest adverse effects on the thermodynamic properties of non-anticoagulated serum components (Figure 6b, Table 9).

The treatment with 40 mM Iomeprol, Iobitridol and Iodixanol led to a negligible thermal shift in the hemoglobin of non-anticoagulated erythrocyte samples (peak 1) (Figure 7, Table 10). However, the deconvolved minor transition peaks might be attributed to the helical erythrocyte component (peak 2) and transferrin (peak 3). These peaks drastically fell in the *T_max_* (~5–6 °C decrease) in the presence of each contrast medium (Figure 7, Table 10). The enthalpy of the major HSA transition (peak 1) decreased considerably (19%) when 40 mM Iodixanol was applied. 

In contrast to the results found in the serum samples, the treatments with each contrast agent considerably increased the enthalpy change of non-anticoagulated hemoglobin transition (peak 1) (30%, 33% and 22%) (Figure 8a, Table 11). These data suggest a putative interaction between the contrast media and hemoglobin, which may make the conformation of hemoglobin more rigid. The enthalpy change of the deconvolved transitions decreased the most in the presence of Iomeprol (91%, 93%) and Iobitridol (73%, 80%). This can be attributed to the helical erythrocyte component (peak 2) and transferrin (peak 3). Iodixanol decreased the enthalpy of the helical erythrocyte component (75%), and transferrin (50%) transitions significantly, but not as much as Iomeprol and Iobitridol (Figure 8a, Table 11). In line with this, Iomeprol and Iobitridol reduced the total enthalpy of non-anticoagulated erythrocyte samples more considerably than Iodixanol (Figure 8b, Table 12). Procoagulant effects have been previously observed in the case of non-ionic contrast media [7]; however, a randomized trial has found no significant correlation between procoagulation and non-ionic contrast media [6]. 

Transferrin is essential in maintaining the coagulation balance [56]. Based on our findings, we believe that the drastic change in the thermodynamic properties of transferrin may lead to an inadequate function of the protein, which further supports the procoagulant effects of non-ionic contrast media. Out of the three contrast agents, Iodixanol had the strongest effects on both the thermodynamic stability and the denaturation energy consumption of anticoagulated blood plasma, erythrocyte components and non-anticoagulated serum proteins. Although each contrast medium had a significant adverse impact on the non-anticoagulated erythrocyte components, the effect of Iodixanol was surprisingly less pronounced. The fact that, in some cases, the negative impact of Iodixanol was greater may be due to its dimeric structure, in which 3-3 iodine atoms are bound to each benzene ring. This suggests that six iodine atoms tend to interact with the blood components with higher affinity, compared to Iomeprol and Iobitridol, which only contain three iodine atoms. 

## 5. Conclusions

Our results suggest that out of the three contrast media we used, Iodixanol can produce the most significant impact. However, Iomeprol and Iobitridol can also considerably influence the thermodynamic stability of anticoagulated and non-anticoagulated erythrocyte proteins (hemoglobin, helical erythrocyte component and transferrin). These findings indicate that the applied contrast media may modify protein functions and, therefore, lead to cardiovascular dysfunctions or thrombus formation.

This work clearly points out some weaknesses of current contrast materials and emphasizes the need for developing a new generation of contrast media in order to avoid harmful effects on human blood components.

## Figures and Tables

**Figure 1 diagnostics-13-02523-f001:**
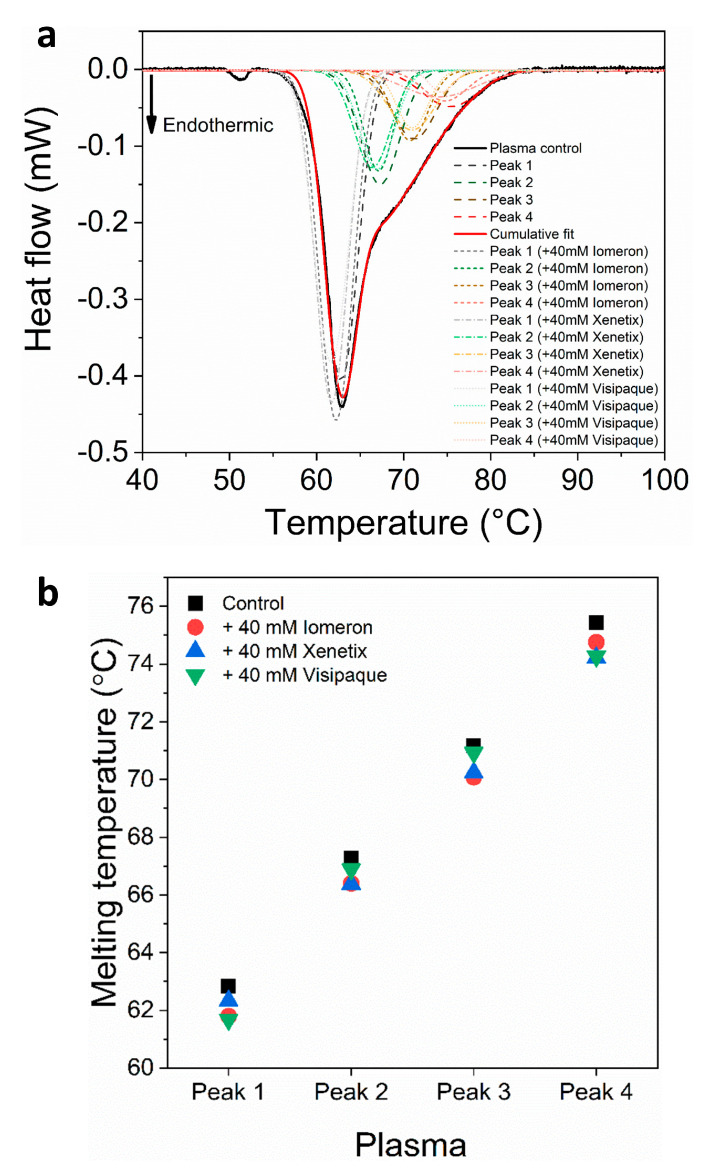
(**a**) Deconvoluted thermal denaturation curves of human blood plasma in the absence (experimental curve—solid black, cumulative fit—solid red and dashed lines) and in the presence of 40 mM Iomeprol, Iobitridol and Iodixanol (short dashed, short dashed dot and short dotted lines), respectively. (**b**) Schematic representation of the deconvoluted thermal transition peaks of anticoagulated plasma as a function of the melting temperature in the absence and presence of 40 mM Iomeron, Xenetix and Visipaque.

**Figure 2 diagnostics-13-02523-f002:**
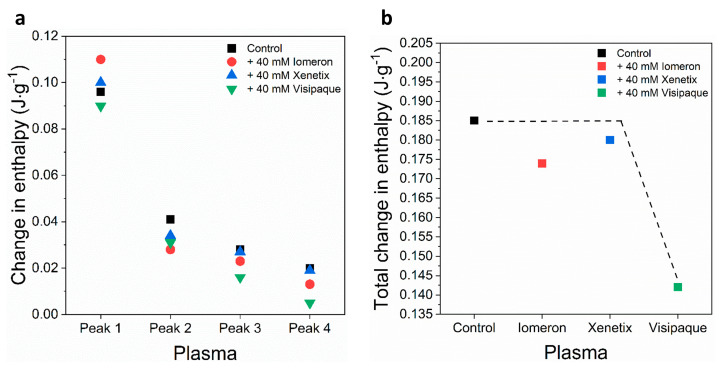
Schematic representation of the deconvoluted thermal transition peak of anticoagulated plasma as a function of the (**a**) calculated enthalpy change and (**b**) total enthalpy change in the absence and presence of 40 mM Iomeron, Xenetix and Visipaque. The black dotted line represents the highest total change in enthalpy.

**Figure 3 diagnostics-13-02523-f003:**
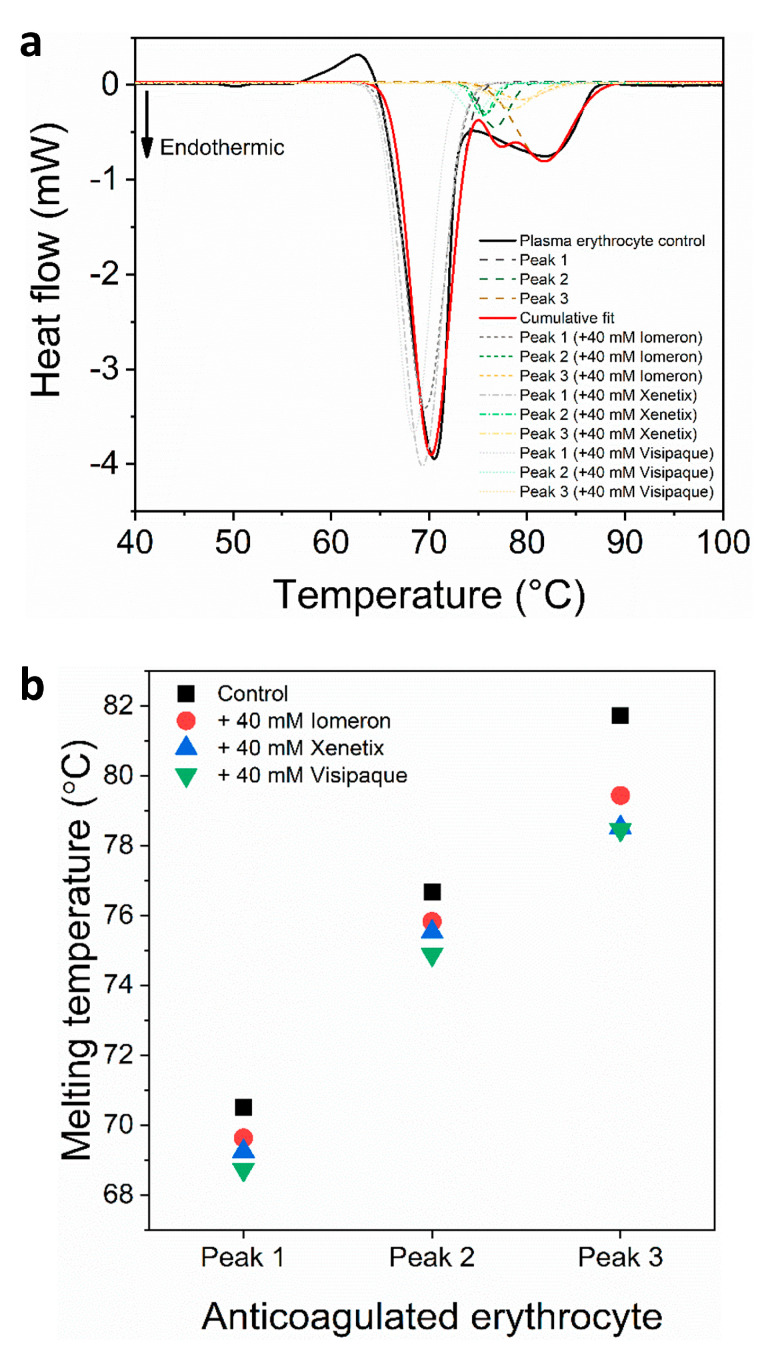
(**a**) Deconvoluted thermal denaturation curves of human erythrocytes of anticoagulated blood in the absence (experimental curve—solid black, cumulative fit—solid red and dashed lines) and in the presence of 40 mM Iomeprol, Iobitridol and Iodixanol (short dashed, short dashed dot and short dotted lines), respectively. (**b**) Schematic representation of the deconvoluted thermal transition peaks of anticoagulated erythrocyte as a function of the melting temperature in the absence and presence of 40 mM Iomeron, Xenetix and Visipaque.

**Figure 4 diagnostics-13-02523-f004:**
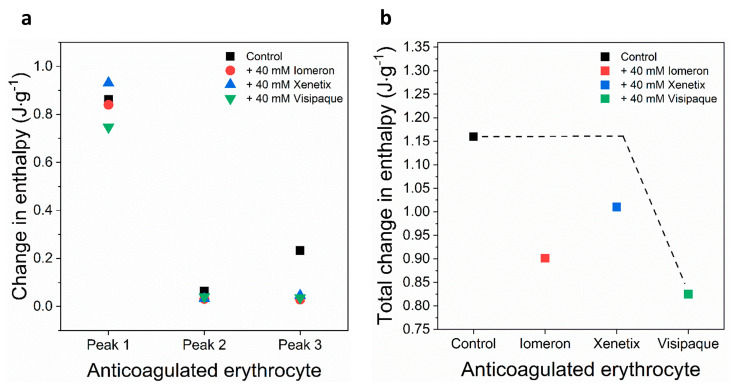
Schematic representation of the deconvoluted thermal transition peaks of anticoagulated erythrocyte as a function of the (**a**) calculated enthalpy change and (**b**) total enthalpy change in the absence and presence of 40 mM Iomeron, Xenetix and Visipaque. The black dotted line represents the highest total change in enthalpy.

**Figure 5 diagnostics-13-02523-f005:**
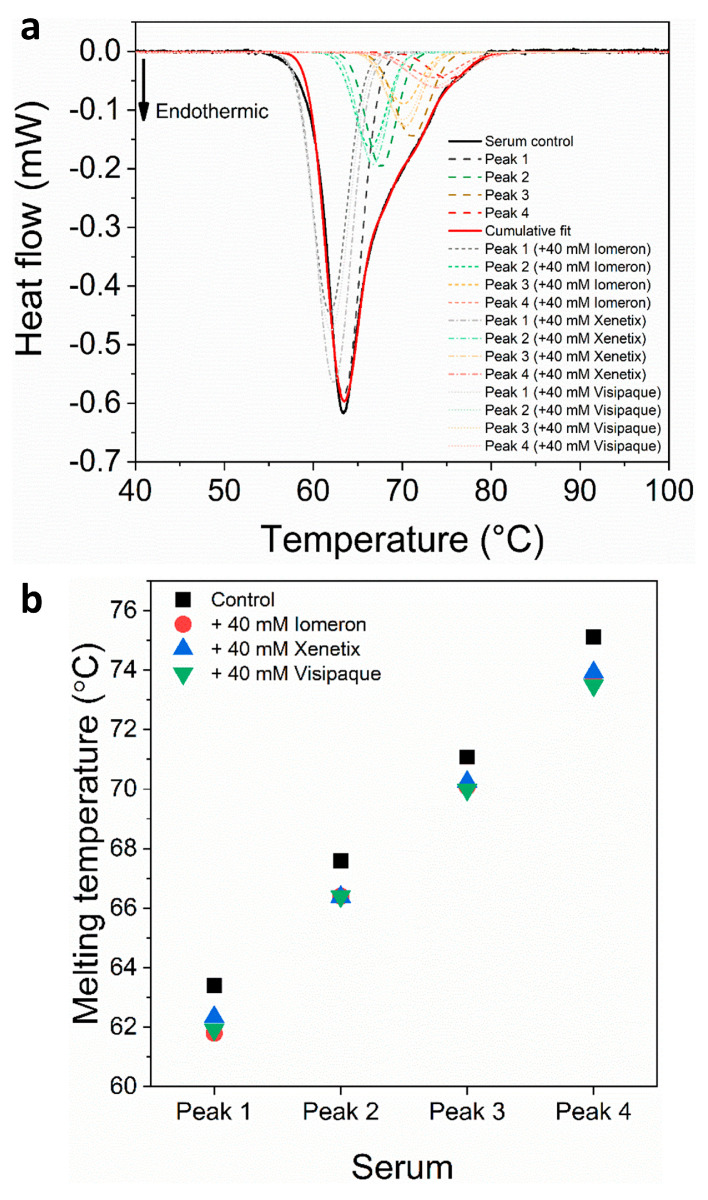
(**a**) Deconvoluted thermal denaturation curves of human blood serum in the absence (experimental curve—solid black, cumulative fit—solid red and dashed lines) and in the presence of 40 mM Iomeprol, Iobitridol and Iodixanol (short dashed, short dashed dot and short dotted lines), respectively. (**b**) Schematic representation of the deconvoluted thermal transition peaks of non-anticoagulated serum as a function of the melting temperature in the absence and presence of 40 mM Iomeron, Xenetix and Visipaque.

**Figure 6 diagnostics-13-02523-f006:**
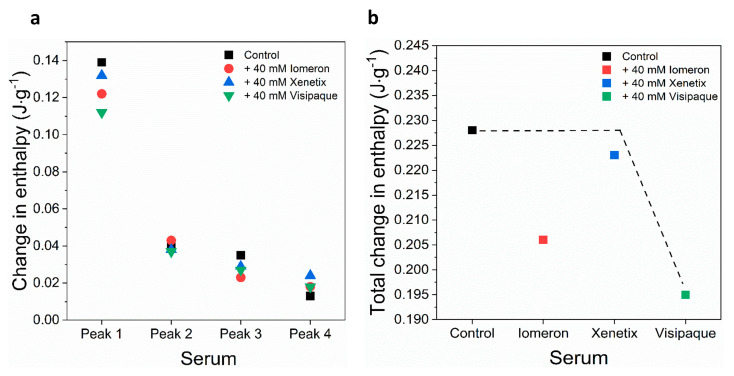
Schematic representation of the deconvoluted thermal transition peaks of non-anticoagulated serum as a function of the (**a**) calculated enthalpy change and (**b**) total change in enthalpy in the absence and presence of 40 mM Iomeron, Xenetix and Visipaque. The black dotted line represents the highest total change in enthalpy.

**Figure 7 diagnostics-13-02523-f007:**
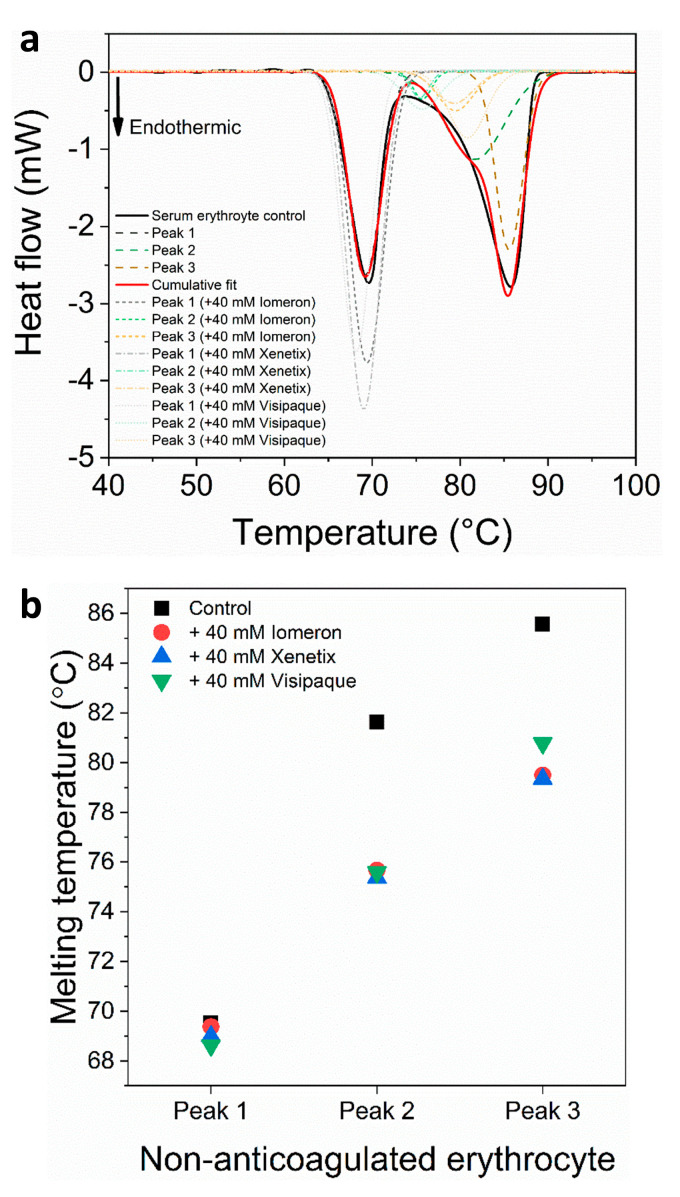
(**a**) Deconvoluted thermal denaturation curves of human erythrocyte of non-anticoagulated blood in the absence (experimental curve—solid black, cumulative fit—solid red and dashed lines) and in the presence of 40 mM Iomeprol, Iobitridol and Iodixanol (short dashed, short dashed dot and short dotted lines), respectively. (**b**) Schematic representation of the deconvoluted thermal transition peaks of non-anticoagulated erythrocyte as a function of the melting temperature in the absence and presence of 40 mM Iomeron, Xenetix and Visipaque.

**Figure 8 diagnostics-13-02523-f008:**
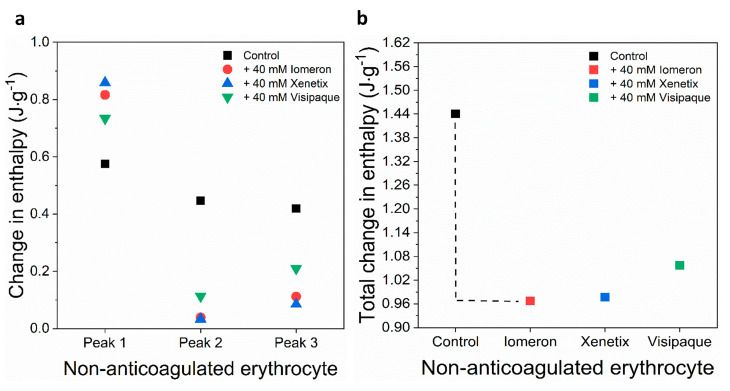
Schematic representation of the deconvoluted thermal transition peaks of non-anticoagulated erythrocyte as a function of the (**a**) calculated enthalpy change and (**b**) total change in enthalpy in the absence and presence of 40 mM Iomeron, Xenetix and Visipaque. The black dotted line represents the highest total change in enthalpy.

**Table 1 diagnostics-13-02523-t001:** The melting temperatures of the deconvoluted curves of anticoagulated plasma in the absence and presence of 40 mM Iomeprol, Xenetix and Visipaque. The abbreviations correspond to the melting temperatures of the control (*T_max1–4(c)_*), Iomeprol (*T_max1–4(i)_*), Xenetix (*T_max1–4(x)_*) and Visipaque (*T_max1–4(v)_*) treated samples.

Blood	Melting Temperature of Anticoagulated Plasma (°C)															
	Control				Iomeron				Xenetix			Visipaque				
	*T_max1(c)_*	*T_max2(c)_*	*T_max3(c)_*	*T_max4(c)_*	*T_max1(i)_*	*T_max2(i)_*	*T_max3(i)_*	*T_max4(i)_*	*T_max1(x)_*	*T_max2(x)_*	*T_max3(x)_*	*T_max4(x)_*	*T_max1(v)_*	*T_max2(v)_*	*T_max3(v)_*	*T_max4(v)_*
	62.9	67.3	71.2	75.4	61.8	66.4	70.1	74.8	62.3	66.4	70.2	74.2	61.7	66.9	70.9	74.3

**Table 2 diagnostics-13-02523-t002:** Change in the enthalpy of the deconvoluted DSC plots of anticoagulated plasma in the absence and presence of 40 mM Iomeprol, Xenetix and Visipaque. The abbreviations correspond to the enthalpy changes of the control (*ΔH*_1-4(c)_), Iomeprol (*ΔH_1–4(i)_*), Xenetix (*ΔH_1–4(x)_*) and Visipaque (*ΔH_1–4(v)_*) treated samples.

Blood	Change in Enthalpy of Anticoagulated Plasma (J·g^−1^)															
	Control				Iomeron				Xenetix				Visipaque			
	*Δ* *H_1(c)_*	*Δ* *H_2(c)_*	*Δ* *H_3(c)_*	*Δ* *H_4(c)_*	*Δ* *H_1(i)_*	*Δ* *H_2(i)_*	*Δ* *H_3(i)_*	*Δ* *H_4(i)_*	*Δ* *H_1(x)_*	*Δ* *H_2(x)_*	*Δ* *H_3(x)_*	*Δ* *H_4(x)_*	*Δ* *H_1(v)_*	*Δ* *H_2(v)_*	*Δ* *H_3(v)_*	*Δ* *H_4(v)_*
	0.09	0.04	0.02	0.02	0.11	0.02	0.02	0.01	0.10	0.03	0.02	0.01	0.09	0.03	0.01	0.005

**Table 3 diagnostics-13-02523-t003:** Summary of the total enthalpy change of the deconvoluted thermal transition curves of anticoagulated plasma in the absence and presence of 40 mM Iomeprol, Xenetix and Visipaque.

Blood	Change in Total Enthalpy of Anticoagulated Plasma (J·g^−1^)			
	Control	Iomeron	Xenetix	Visipaque
	*Δ* *H_total(c)_*	*Δ* *H_total(i)_*	*Δ* *H_total(x)_*	*Δ* *H_total(v)_*
	0.18	0.17	0.18	0.14

**Table 4 diagnostics-13-02523-t004:** The melting temperatures of the deconvoluted curves of anticoagulated erythrocyte components in the absence and presence of 40 mM Iomeprol, Xenetix and Visipaque. The abbreviations correspond to the melting temperatures of the control (*T_max1–3(c)_*), Iomeprol (*T_max1–3(i)_*), Xenetix (*T_max1–3(x)_*) and Visipaque (*T_max1–3(v)_*) treated samples.

Blood	Melting Temperature of Anticoagulated Erythrocyte (°C)											
	Control			Iomeron			Xenetix			Visipaque		
	*T_max1(c)_*	*T_max2(c)_*	*T_max3(c)_*	*T_max1(i)_*	*T_max2(i)_*	*T_max3(i)_*	*T_max1(x)_*	*T_max2(x)_*	*T_max3(x)_*	*T_max1(v)_*	*T_max2(v)_*	*T_max3(v)_*
	70.5	76.7	81.7	69.6	75.8	79.4	69.3	75.5	78.5	68.7	74.9	78.5

**Table 5 diagnostics-13-02523-t005:** Change in the enthalpy of the deconvoluted DSC plots of anticoagulated blood components in the absence and presence of 40 mM Iomeprol, Xenetix and Visipaque. The abbreviations correspond to the enthalpy changes of the control (*ΔH_1–3(c)_*), Iomeprol (*ΔH_1–3(i)_*), Xenetix (*ΔH_1–3(x)_*) and Visipaque (*ΔH_1–3(v)_*) treated samples.

Blood	Change in Enthalpy of Anticoagulated Erythrocyte (J·g^−1^)											
	Control			Iomeron			Xenetix			Visipaque		
	*Δ* *H1(c)*	*Δ* *H_2(c)_*	*Δ* *H_3(c)_*	*Δ* *H_1(i)_*	*Δ* *H_2(i)_*	*Δ* *H_3(i)_*	*Δ* *H_1(x)_*	*Δ* *H_2(x)_*	*Δ* *H_3(x)_*	*Δ* *H_1(v)_*	*Δ* *H_2(v)_*	*Δ* *H_3(v)_*
	0.86	0.06	0.23	0.84	0.03	0.02	0.93	0.03	0.04	0.74	0.04	0.03

**Table 6 diagnostics-13-02523-t006:** Summary of the total enthalpy change of the deconvoluted thermal transition curves of anticoagulated erythroctyte in the absence and presence of 40 mM Iomeprol, Xenetix and Visipaque.

Blood	Change in Total Enthalpy of Anticoagulated Erythrocyte (J·g^−1^)			
	Control	Iomeron	Xenetix	Visipaque
	*Δ* *H_total(c)_*	*Δ* *H_total(i)_*	*Δ* *H_total(x)_*	*Δ* *H_total(v)_*
	1.16	0.90	1.01	0.82

**Table 7 diagnostics-13-02523-t007:** The melting temperatures of the deconvoluted curves of non-anticoagulated serum in the absence and presence of 40 mM Iomeprol, Xenetix and Visipaque. The abbreviations correspond to the melting temperatures of the control (*T_max1–4(c)_*), Iomeprol (*T_max1–4(i)_*), Xenetix (*T_max1–4(x)_*) and Visipaque (*T_max1–4(v)_*) treated samples.

Blood	Melting Temperature of Non-Anticoagulated Serum (°C)															
	Control				Iomeron				Xenetix				Visipaque			
	*T_max1(c)_*	*T_max2(c)_*	*T_max3(c)_*	*T_max4(c)_*	*T_max1(i)_*	*T_max2(i)_*	*T_max3(i)_*	*T_max4(i)_*	*T_max1(x)_*	*T_max2(x)_*	*T_max3(x)_*	*T_max4(x)_*	*T_max1(v)_*	*T_max2(v)_*	*T_max3(v)_*	*T_max4(v)_*
	63.4	67.6	71.1	75.1	61.8	66.4	70.1	73.7	62.3	66.4	70.2	73.9	61.9	66.4	70.0	73.5

**Table 8 diagnostics-13-02523-t008:** Change in the enthalpy of the deconvoluted DSC plots of non-anticoagulated serum in the absence and presence of 40 mM Iomeprol, Xenetix and Visipaque. The abbreviations correspond to the enthalpy changes of the control (*ΔH_1–4(c)_*), Iomeprol (*ΔH_1–4(i)_*), Xenetix (*ΔH_1–4(x)_*) and Visipaque (*ΔH_1–4(v)_*) treated samples.

Blood	Change in Enthalpy of Non-Anticoagulated Serum (J·g^−1^)															
	Control				Iomeron				Xenetix				Visipaque			
	*Δ* *H_1(c)_*	*Δ* *H_2(c)_*	*Δ* *H_3(c)_*	*Δ* *H_4(c)_*	*Δ* *H_1(i)_*	*Δ* *H_2(i)_*	*Δ* *H_3(i)_*	*Δ* *H_4(i)_*	*Δ* *H_1(x)_*	*Δ* *H_2(x)_*	*Δ* *H_3(x)_*	*Δ* *H_4(x)_*	*Δ* *H_1(v)_*	*Δ* *H_2(v)_*	*Δ* *H_3(v)_*	*Δ* *H_4(v)_*
	0.13	0.04	0.03	0.01	0.12	0.04	0.02	0.01	0.13	0.03	0.02	0.02	0.11	0.03	0.02	0.01

**Table 9 diagnostics-13-02523-t009:** Summary of the total enthalpy change of the deconvoluted thermal transition curves of non-anticoagulated serum in the absence and presence of 40 mM Iomeprol, Xenetix and Visipaque.

Blood	Change in Total Enthalpy of Non-Anticoagulated Serum (J·g^−1^)			
	Control	Iomeron	Xenetix	Visipaque
	*Δ* *H_total(c)_*	*Δ* *H_total(i)_*	*Δ* *H_total(x)_*	*Δ* *H_total(v)_*
S	0.22	0.20	0.22	0.19

**Table 10 diagnostics-13-02523-t010:** The melting temperatures of the deconvoluted curves of non-anticoagulated erythrocytes in the absence and presence of 40 mM Iomeprol, Xenetix and Visipaque. The abbreviations correspond to the melting temperatures of the control (*T_max1–3(c)_*), Iomeprol (*T_max1–3(i)_*), Xenetix (*T_max1–3(x)_*) and Visipaque (*T_max1–3(v)_*) treated samples.

Blood	Melting Temperature of Non-Anticoagulated Erythrocyte (°C)											
	Control			Iomeron			Xenetix			Visipaque		
	*T_max1(c)_*	*T_max2(c)_*	*T_max3(c)_*	*T_max1(i)_*	*T_max2(i)_*	*T_max3(i)_*	*T_max1(x)_*	*T_max2(x)_*	*T_max3(x)_*	*T_max1(v)_*	*T_max2(v)_*	*T_max3(v)_*
	69.6	81.6	85.6	69.4	75.7	79.5	69	75.4	79.4	68.6	75.6	80.8

**Table 11 diagnostics-13-02523-t011:** Change in the enthalpy of the deconvoluted DSC plots of non-anticoagulated erythrocyte in the absence and presence of 40 mM Iomeprol, Xenetix and Visipaque. The abbreviations correspond to the enthalpy changes of the control (*ΔH_1–3(c)_*), Iomeprol (*ΔH_1–3(i)_*), Xenetix (*ΔH_1–3(x)_*) and Visipaque (*ΔH_1–3(v)_*) treated samples.

Blood	Change in Enthalpy of Non-Anticoagulated Erythrocyte (J·g^−1^)											
	Control			Iomeron			Xenetix			Visipaque		
	*Δ* *H_1(c)_*	*Δ* *H_2(c)_*	*Δ* *H_3(c)_*	*Δ* *H_1(i)_*	*Δ* *H_2(i)_*	*Δ* *H_3(i)_*	*Δ* *H_1(x)_*	*Δ* *H_2(x)_*	*Δ* *H_3(x)_*	*Δ* *H_1(v)_*	*Δ* *H_2(v)_*	*Δ* *H_3(v)_*
	0.57	0.44	0.41	0.81	0.03	0.11	0.85	0.03	0.08	0.73	0.11	0.21

**Table 12 diagnostics-13-02523-t012:** Summary of the total enthalpy change of the deconvoluted thermal transition curves of non-anticoagulated erythrocyte in the absence and presence of 40 mM Iomeprol, Xenetix and Visipaque.

Blood	Change in Total Enthalpy of Non-Anticoagulated Erythrocyte (J·g^−1^)			
	Control	Iomeron	Xenetix	Visipaque
	*Δ* *H_total(c)_*	*Δ* *H_total(i)_*	*Δ* *H_total(x)_*	*Δ* *H_total(v)_*
	1.44	0.96	0.97	1.05

## Data Availability

Data are contained within the article.

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
