# Peer review of "Deconvolution Analysis of the Non-Ionic Iomeprol, Iobitridol and Iodixanol Contrast Media-Treated Human Whole Blood Thermograms: A Comparative Study"

_diagnostics, 2023, doi:10.3390/diagnostics13152523_

Round 1

Reviewer 1 Report

This work presents the results of experimental studies of blood samples affected by different non-ionic iodine-based contrast agents. These substances may affect the stability of blood constituents, and applying differential scanning calorimetric is an adequate method to clarify this issue. The material, method and experimental approach are described in detail, route and procedure convince in the obtained results, which are valuable to characterize the safety of these substances for usage in medical diagnostics as contrast agents.

 Thus, I recommend accepting this work after minor corrections to some points of description:

1)     some estimations of the uncertainty of determining the phase transition temperature would be requested: Table 1 reports these temperatures with accuracy up to 0.01 K which may be too detailed, it rather expected 0.1 K taking into account typical procedures of finding maxima for the DSC curves of biological objects (especially after applying any decomposition procedure).

2)     The same can improve the discussion of Figs. 5-6: although there is already textual decryption when one can neglect by deviations between closely placed markers, some estimations of numerical uncertainty ranges can be more demonstrative.

Author Response

Dear Reviewer 1, 

Reviewer 2 Report

2023-07-14

The review of the submission diagnostics-2498556.

After careful examination of the above-mentioned document, I’d like to state, that the authors did not place the figures in the common to MDPI place after first mention. The Figure 1 appeared after Figure 7 is mentioned. The authors need to rearrange the submission in MDPI way, otherwise it is really hard to read.

More specific comments below:

1. The submission is devoted to a very important issue - deconvolution of DSC curves of blood samples. By itself it is a really nice paper, written in proper English.
2. The issue is really important for blood analysis. The paper itself is average.
3. It is really hard to say what it adds to the subject area compared with other published material, since it is very inconvenient to read.
4. Authors need to distribute Figures according to the common MDPI rules. After that it will be possible to evaluate the quality.
5. Since the Figures are spatially separated from their description, it is hard to say whether conclusions are consistent with the evidence and arguments presented.
6. References are appropriate. 7. Authors need to distribute both Figures and Tables according to the common MDPI rules to make the reviewer's work easy. After that it will be possible to evaluate the quality.

Good luck

Author Response

Dear Reviewer 2, 

please, see the attachment.

Round 2

Reviewer 2 Report

As for me, It looks acceptable. All corrections are relevant and hadn't made the submission worse.